



# Two new early instrumental records of air pressure and temperature for the southern European Alps

Yuri Brugnara[1,2], Michael Horn[3], and Isabella Salvador[4]

[1]Oeschger Centre for Climate Change Research, University of Bern, Switzerland
[2]Institute of Geography, University of Bern, Switzerland
[3]University Library, University of Bern, Switzerland
[4]MUSE Science Museum, Trento, Italy

**Correspondence:** Yuri Brugnara (yuri.brugnara@giub.unibe.ch)

**Abstract.** Central Europe is among the regions with the largest availability of pre-industrial meteorological records. In the Alps, however, such records are relatively rare, especially in the southern slope. We recently found and digitized two new pressure and temperature series for the Alpine cities of Rovereto (1800–1839) and Bolzano/Bozen (1842–1849) covering together the first half of the 19th century, a period characterized by large climate variability and important extreme events. The
meteorological record of Rovereto, in particular, is the oldest available for the southeastern Alps. We used the shorter record of Bolzano/Bozen as a testbed for different digitization methods, namely citizen science and machine-learning based Optical Character Recognition. The data are converted to modern units, quality controlled, and homogenized. We also provide daily and monthly means together with an estimation of their uncertainty.

## 1 Introduction

Natural climate forcing can have significant impacts on global and regional climate. The most prominent example in modern history is the early 19th century, when two large volcanic eruptions in 1808 and 1815 contributed to several years of unusually low temperatures, particularly in the Eurasian continent (Brönnimann et al., 2019b; Reichen et al., 2022).

Currently available instrumental measurements are often insufficient for a detailed analysis of climate variability before 1850, even in Europe, where many data remain unexploited because they are only available in paper form (Brönnimann et al.,
2019a). Unfortunately, digitization is a very expensive process and the quality of early instrumental measurements is also a concern.

Being located at the center of Europe, the Alps were visited and studied by early climatologists already during the Enlightenment. By the early 19th century several meteorological observatories had been established, some of which delivered continuous records until the present day (Auer et al., 2007; Pfister et al., 2019; Brugnara, 2020). This development did not play out equally
over the entire Alpine region, being much slower in the southern slope. In northern Italy, many early records are available from the cities in the Po Plain (e.g., Maugeri et al., 2002; Camuffo et al., 2006, 2017), but hardly any exists from the southern Alpine valleys, especially at daily resolution.

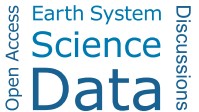

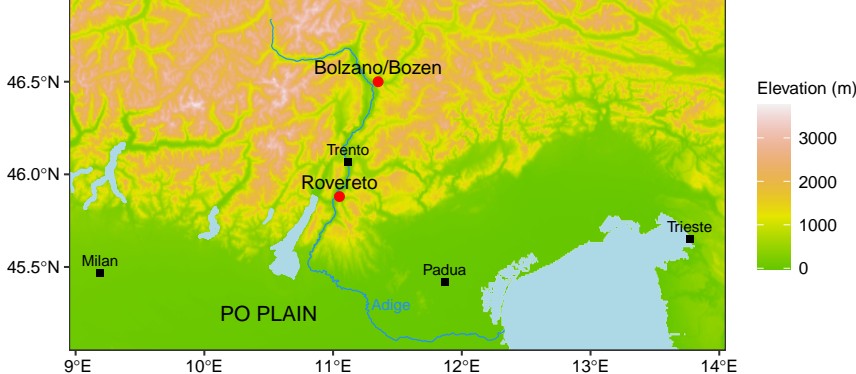

**Figure 1.** Topographic map of the southeastern European Alps showing the positions of Rovereto and Bolzano/Bozen in the Adige Valley. Digital Elevation Model from Jarvis et al. (2008).

Recently we digitized two previously unknown records from the Adige Valley in the southeastern Alps, covering the first half of the 19th century. They include daily instrumental pressure and temperature measurements and were recorded by two science teachers in Rovereto and in Bolzano/Bozen. An additional temperature record from the largest city in the Adige Valley, Trento, is already available to the scientific community, although only at monthly resolution (Auer et al., 2007). The new record of Rovereto begins 16 years earlier, in 1800, and is thus the oldest instrumental record currently available for the southeastern Alps, as well as the only one covering the cold period in the 1810s.

Early instrumental climate data are usually digitized through manual keying. In recent years, however, alternative digitization approaches such as automated Optical Character Recognition (OCR) and citizen science have become more flexible and accessible to scientists. In this article we also briefly discuss our experience with these approaches.

## 2 Data Sources

### 2.1 Rovereto

Rovereto (204 m above mean sea level) is the second largest city of the Italian Province of Trento (also known as Trentino) and historically an important cultural centre in the region. It currently hosts a Centennial Observing Station (dating back to 1882) recognized by the World Meteorological Organization (see Brugnara et al., 2016).

The record that we recovered is by the abbot Giuseppe Bonfioli (1754–1839) and covers the period from January 1800 to August 1839, with irregular measurements back to 1782. Unfortunately, not much is known about the measurements and their author. We only know that Bonfioli was a teacher of experimental physics and from 1812 was enrolled in the "Accademia Roveretana degli Agiati", a learned society based in Rovereto. The original data sheets (Fig. 2) are stored at the Biblioteca Civica (public library) in Rovereto (Archive Reference: Ms. 57. 6) together with several manuscripts about various fields of physics.

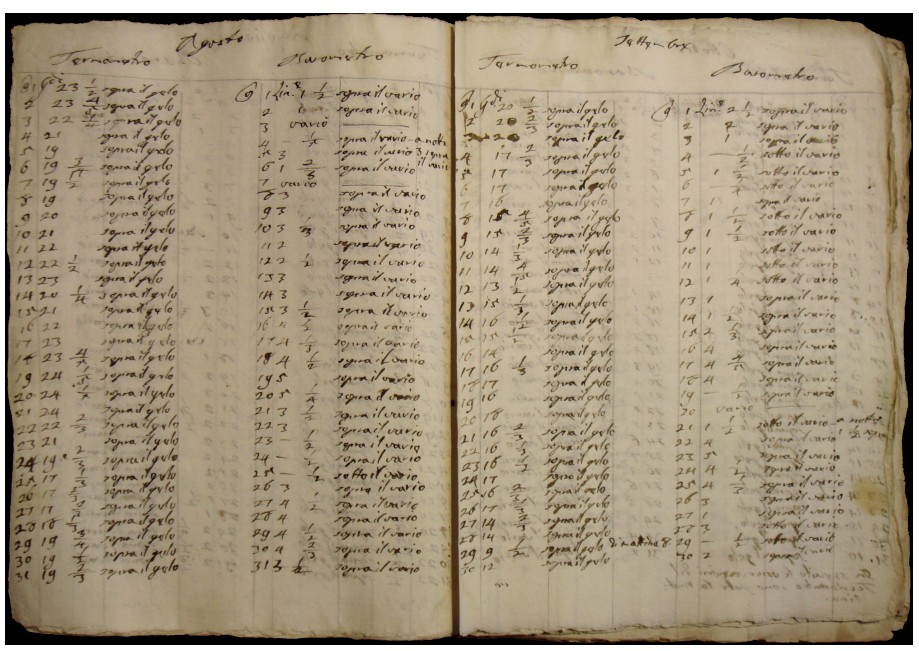

**Figure 2.** Extract of Bonfioli's weather diary (August and September 1809). Source: Archivio Biblioteca Civica Rovereto, Ms. 57.6.

The only information about the measurements were published by one of Bonfioli's peers, the Austrian meteorologist Franz Zallinger zum Thurn (1743–1828), who had asked Bonfioli for data to compare with his own measurements in Innsbruck (Zallinger zum Thurn, 1809). Initially Bonfioli measured only once a day – at sunrise in winter and 1 hour before sunset in summer according to Zallinger zum Thurn (1809) – with few exceptions. The position of the thermometer also changed, being exposed in the shadow to the northwest in winter and to the west in summer (presumably with respect to his apartment). The transition between the two configurations was not on a fixed date but was noted down in the data sheets. From 1816 onward there are usually two temperature measurements per day, although we do not know what the observation times were or where the thermometer was hung. It seems unlikely that it would have been moved every day, though.

The barometer used by Bonfioli was self made, with a diameter of 4.5 mm at the tube and 45 mm at the cistern. The readings are given as deviations from a zero level of 27.5 Paris inches (744.4 mmHg). There is usually only one pressure measurement per day throughout the record, which we assume was made in the morning except during the first year, when lower pressure levels in summer are compatible with afternoon observations.

In addition to the measurements, Bonfioli wrote down detailed accounts of the main weather events for most years starting from 1778 (see Salvador et al., 2017, 2018). Zallinger zum Thurn (1809) also mentions a 10-year pressure record for Rovereto by the abbot Francesco Ferrari (1767–1821), of which we could not find any trace. We did find and digitize, however, a short pressure and temperature record for the years 1827–1829 by an anonymous observer (published by the newspaper *Il Messagiere Tirolese*), which we use for validation purposes.





| 1842 Jänner | Barometerstand im Par. M. reduzirt auf 0 Reaumur | | | | | | Thermometerstand in Graden nach Reaumur | | | Witterung | | |
|---|---|---|---|---|---|---|---|---|---|---|---|---|
| | 7 Uhr Morg. | | 1 Uhr Nachm. | | 9 Uhr Abends | | 7 Uhr Morg. | 1 Uhr Nachm. | 9 Uhr Abends | Morgens | Nachmittags | Abends |
| | Zoll | Linien | Zoll | Linien | Zoll | Linien | | | | | | |
| 14 | 27 | 4 | 27 | 4 | 27 | 4 | − 3 | 0 | − 2 | trüb | trüb | Schnee |
| 15 | 27 | 6 | 27 | 7 | 27 | 6 | − 2 | + 1 | + 1 | detto | heiter | heiter |
| 16 | 27 | 6 | 27 | 7 | 27 | 6 | 0 | + 2 | + 2 | Nebel | Wolken | Regen |
| 17 | 27 | 6 | 27 | 6 | 27 | 5 | + 2 | + 3 | + 2 | Regen | trüb | detto |
| 18 | 27 | 3 | 27 | 4 | 27 | 3 | + 1 | + 4 | + 2 | Nebel | detto | trüb |
| 19 | 27 | 2 | 27 | — | 27 | — | + 3 | + 6 | + 4 | trüb | heiter | Wolken |
| 20 | 27 | — | 27 | — | 27 | 1 | + 3 | + 5 | + 2 | detto | trüb | heiter |

**Figure 3.** Example of a data table published on the *Bozner Wochenblatt*. Source: Landesbibliothek Dr. Friedrich Teßmann (https://digital. tessmann.it, last access 14 April 2022). License: CC BY-NC-ND 4.0 (https://creativecommons.org/licenses/by-nc-nd/4.0/).

## 2.2 Bolzano/Bozen

Bolzano (or Bozen; 262 m above mean sea level) is the capital of the homonymous Province (also known as South Tyrol) and lies about 70 km north-northeast of Rovereto (Fig. 1). Monthly temperature data for Bolzano are available in public data sets starting from 1850 (Auer et al., 2007). We digitized sub-daily data (three measurements per day) for the period 1842–1849.

The data are from the *Bozner Wochenblatt*, a newspaper published in Bolzano. They were measured at the Franciscan cloister in the city center, probably by the gymnasium's physics teacher Cyrill Conzin (1816–1868), and were published in weekly tables (Fig. 3). The data had already been used in the late 1870s by the successor of Conzin, August Pölt (Pölt, 1879, 1880), but had been forgotten since.

In addition to pressure and temperature we also digitized the observed weather conditions, which are always described using one of the following terms: clear, clouds, overcast, rain, snow, fog, storm, thunderstorm. The *Bozner Wochenblatt* (later *Bozner Zeitung*) continued to publish the data tables until 1873, so that additional 24 years remain to be digitized.

Unfortunately, we could not find any relevant information about the instruments and exposure. The data have very coarse resolution, perhaps because of printing constraints, but pressure is explicitly reduced to $0°C$. According to Pölt (1879) the original data sheets do not exist anymore.

It might be possible to find even older instrumental data for Bolzano. For instance, the brother of Franz Zallinger zum Thurn measured pressure, temperature, and humidity in Bolzano at least during the years 1783–1784, for which monthly means have been published (Zallinger zum Thurn, 1784); and Pölt (1879) mentions pressure and temperature measurements made between 1830–1838 by two foresters.

## 2.3 Validation Data

We compare our data with temperature and pressure series from Milan (Maugeri et al., 2002), Padua (Cocheo and Camuffo, 2002), Trento (Auer et al., 2007), Trieste (Raicich and Colucci, 2021), and from the nearest grid point of the Twentieth Century Reanalysis (20CR) version 3 (Slivinski et al., 2019) and the paleo-reanalysis EKF400 version 2 (Franke et al., 2020). We also use the ensemble mean of EKF400 as reference to homogenize the record of Rovereto.



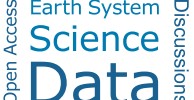

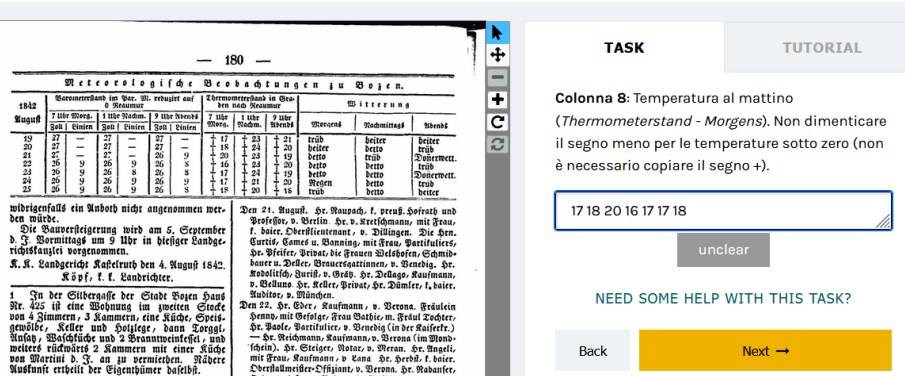

**Figure 4.** Citizen science interface on Zooniverse for the Bolzano data. Image source: Landesbibliothek Dr. Friedrich Teßmann (https://digital.tessmann.it, last access 14 April 2022). License: CC BY-NC-ND 4.0 (https://creativecommons.org/licenses/by-nc-nd/4.0/).

## 3 Methods

### 3.1 Digitization

85 The record of Rovereto was keyed by experienced geography students of the University of Bern. The record of Bolzano, on the other hand, was digitized three times using different methods: 1) manual keying by the authors, 2) manual keying by volunteers through the citizen science portal Zooniverse, and 3) using a self-developed OCR software. The aim was to compare the quality of digitization by citizen scientists and OCR. After a quality control, the self-keyed data were used for validation assuming that they had no errors. Note that typos in the source were not corrected at this stage.

90 ### 3.1.1 Citizen Science

The portal Zooniverse (https://www.zooniverse.org) provides a free and easy-to-use platform to publish a citizen science project on the web. Given that ours was a small project with no formal funding, it could not be given the status of official Zooniverse project (which would widen the pool of volunteers by making it searchable from the Zooniverse home page). The only publicity was through private accounts on social media and at public scientific events. For this reason we expect our results to be overly

95 optimistic, because our pool of volunteers was arguably biased towards well educated young adults.

The volunteers were asked to copy one table at a time, column by column (Fig. 4). For the non-numeric fields, typos were avoided by providing a list of possible values from which volunteers could choose. Each table was digitized by only one volunteer without repetition. Due to the lack of publicity, only about half of the data tables between 1842–1849 could be digitized through citizen science over the course of the project (ca. one year). However, this was sufficient for our aims.



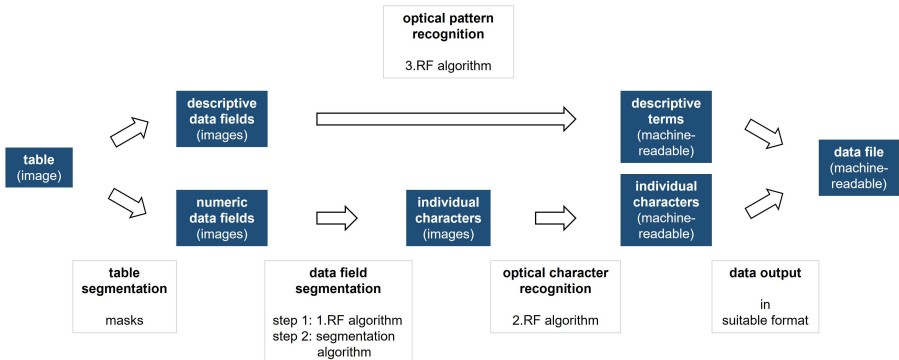

**Figure 5.** Workflow of the OCR-based digitization approach (RF: Random Forest).

### 3.1.2 OCR

The workflow of the OCR-based digitization is summarized in Fig. 5. It begins with the segmentation of the printed data tables with a system of automatically adjustable masks, resulting in a set of individual data fields. The masks cover the entire table with exception of the data fields. Two types of data fields are extracted: numeric data fields containing air pressure and temperature data and descriptive data fields containing descriptive weather conditions. Numeric data fields are further segmented to individual characters: a random forest (RF) algorithm determines the amount of characters in each numeric data field, while a deterministic segmentation algorithm subsequently divides each data field into individual characters, which are then routed to OCR. A second RF algorithm predicts the label for each character (possible labels are: +, -, 0, 1, 2, 3, 4, 5, 6, 7, 8, 9), resulting in machine-readable individual characters. Descriptive data fields are directly routed to optical pattern recognition. A third RF algorithm predicts the label for each pattern, resulting in machine-readable descriptive terms. The data were finally aggregated into a comma-separated flat file.

### 3.2 Unit Conversion and Pressure Corrections

The pressure readings for Bolzano were expressed in Paris inches and tenths of inches (1 Paris inch equals 27.07 $\mathrm{mmHg}$), those of Rovereto in Paris lines (i.e., twelfths of inch). Pressure values are converted to $\mathrm{hPa}$ through the hydrostatic equation and additionally corrected for gravity and reduced to mean sea level (Brugnara et al., 2015). Note that the reduction to sea level was done in order to compare the data with other series in this paper and that the published data are not reduced, since we do not know the exact elevation of the barometers (this is particularly relevant in the case of Rovereto, where we do not even know the exact location within the city).

It is never mentioned whether Bonfioli applied the necessary corrections to his barometer readings (in particular, a temperature correction). By comparing his measurements with a modern-day pressure climatology (2011–2021) from the Rovereto station of the regional weather service (Meteotrentino) it is clear that the readings were not corrected for temperature (Fig. 6). The large annual cycle of the pressure bias lets us assume that the barometer was exposed to the outdoor temperature; therefore,


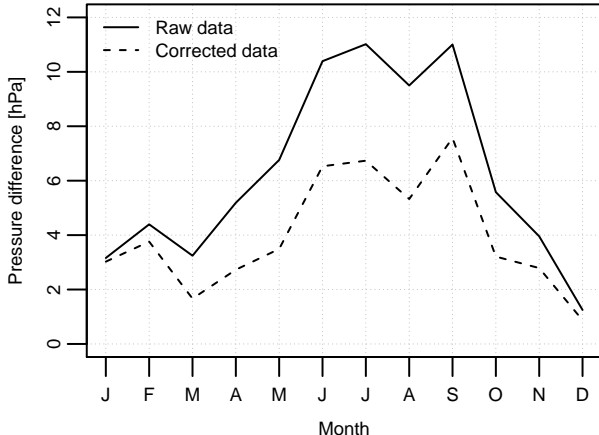

**Figure 6.** Difference between monthly climatological pressure measured by Bonfioli (1800–1839) and by Meteotrentino (2011–2021) before and after applying a temperature correction to the former.

we used the measured temperature to reduce the Rovereto pressure data to $0°C$ (WMO, 2008). Since the pressure measurements are not always made at the same time of the temperature measurements, we could apply the temperature correction only at the level of daily means.

Both temperature records were measured using a mercury thermometer with Réaumur scale. The conversion to Celsius degrees requires multiplying by a factor 1.25. We use the descriptive weather information from Bolzano to check the zero of the temperature scale by looking at the relationship between temperature and the frequency of snowfall.

### 3.3 Observation Times and Daily Means

While observation times are explicitly given for Bolzano (7:00 AM, 1:00 PM, and 9:00 PM local time), they are only labeled
as "morning" and "afternoon" for Rovereto. We know that the measurements were taken around sunrise in winter and 1 hour before sunset in summer, at least until 1808 (see Sect. 2.1). To estimate the exact observation times, we compare the temperature data with the anonymous record of 1827–1829, for which observation times are given explicitly.

The calculation of daily means must take into account the diurnal cycle of pressure and temperature, particularly when only one measurement per day is available as in the case of Rovereto. For temperature, we use the multiple linear regression (MLR)
approach described in Brugnara et al. (2022a). In short, we train a MLR model on modern-day sub-hourly data to reproduce true daily means from only one or two measurements in a day. The model can be expressed as:

$$T_m = a_0 + \sum_{i=1}^{n+2} a_i x_i + \epsilon \tag{1}$$



where the predictors $x_i$ are the elements of the vector

$$\boldsymbol{x} = \left[ sin\left(\frac{2\pi j}{366}\right), cos\left(\frac{2\pi j}{366}\right), T_1(, T_2) \right], \tag{2}$$

$T_m$ is the mean daily temperature (i.e., the predictand), $T_{1/2}$ are the observed temperature values on the analyzed day, $j$ is the Julian day, $n$ is the number of measurements (one or two), $a_i$ are the regression parameters, and $\epsilon$ is the residual error. The first two elements of $\boldsymbol{x}$ are added to capture the seasonal variability of the diurnal cycle. All temperatures are expressed in terms of anomalies.

For pressure, which has a smaller diurnal cycle, we adjust the aritmethic average of the available observations using the
modern-day mean diurnal cycle of the specific month.

For Bolzano there are no publicly available long sub-daily records, therefore we adopted the common approach of taking a weighted average of the three temperature observations with double weight assigned to the evening observation. For pressure we simply take the arithmetic average.

Monthly means are calculated from daily means when no more than four consecutive days are missing (WMO, 2008). Over
99% of the days in the record of Rovereto have at least one measurement, leading to only four missing monthly means for pressure and two for temperature; for Bolzano a few weekly tables where not published, while others were partly unreadable, resulting in three missing monthly means for both pressure and temperature.

## 3.4   Quality Control and Homogeneity

Data quality was checked following the procedure described in Brunet et al. (2020). This includes automatic checks for sta-
tistical outliers, repeated or impossible values, and sudden jumps, as well as visual checks. Values that could not be corrected (i.e., that were not typos) were retained with a quality flag and not used for the calculation of daily means (six instances for Rovereto and 20 for Bolzano).

When necessary we corrected inhomogeneities in the data caused by changes in the measurement procedures. To do so we calculated monthly corrections by comparing our data with the ensemble mean of the EKF400 paleo-reanalysis at the nearest
grid point. The monthly corrections are then transformed into daily corrections by fitting the first two harmonics of a Fourier series (Brugnara et al., 2022a). In order to minimize modifications to the original record, the data are adjusted with respect to the longest homogeneous segment. We detected the points at which inhomogeneities occurred (breakpoints) with the help of the Craddock test (Craddock, 1979).



## 3.5 Error Estimation

We estimate standard errors for each daily and monthly mean value following Brugnara et al. (2022a). Daily errors are given by:

$$E_d = \sqrt{\sum_{i=1}^{4} e_i^2} \tag{3}$$

where $e_1$ is a resolution error, $e_2$ is an error related to the number of measurements in a day, $e_3$ is an error related to the time uncertainty, and $e_4$ is an exposure error (for temperature only). $e_1$ and $e_4$ are in turn defined by the following equations:

$$e_1(\delta, n) = \frac{\delta}{2\sqrt{n}} \tag{4}$$

$$e_4(n, j) = \frac{1}{\sqrt{n}} \left( a + b \sin \frac{2\pi(j - 81)}{N} \right) \tag{5}$$

where $\delta$ is the reporting resolution, $n$ is the number of observations in a day, $a = 0.8$ K, $b = 0.4$ K, and $N$ is the number of days in the year.

For temperature, $e_2$ is the standard deviation of the residuals from Eq. 1 aggregated by month, while $e_3$ is calculated from the modern-day mean diurnal cycle of Rovereto assuming a time uncertainty of 90 minutes (see also Brugnara et al., 2022a). For Bolzano we set $e_2 = e_3 = 0$ given that we do not have modern-day data to calculate those error components (expected to be negligible anyway). Note that systematic errors are not taken into account.

For pressure, $e_2$ is the standard deviation of the differences between true daily means and the average of the available obser-
vations, calculated from the modern-day data and aggregated by month. $e_3$ is calculated in the same way as for temperature. An estimation of $e_4$ for barometer readings is hardly possible, therefore we consider only three error components.

Finally, the standard error for monthly means is:

$$E_m = \frac{1}{N_d} \sqrt{\sum_{i=1}^{N_d} E_{d(i)}^2} \tag{6}$$

where $N_d$ is the number of non-missing daily means in the target month.

## 4 Data Validation and Homogenization

### 4.1 Observation Times in Rovereto

Since we do not know the exact observation times followed by Bonfioli, we tried to estimate them by comparing Bonfioli's temperature measurements with those of an anonymous observer who measured in Rovereto between 1827–1829, with the



aid of the modern-day climatological diurnal cycle. For that we calculate the mean diurnal cycle for each Julian day from
a 15-day window centered on the target day. We then extract the expected mean temperature difference between possible
observation times followed by Bonfioli and the observation times followed by the anonymous observer and compare them with
the measured differences during 1827–1829 (Fig. 7).

The anonymous observer measured before sunrise (between 5:00 AM in summer and 7:00 AM in winter) and at 2:00
PM. The large seasonal cycle in the differences for the morning observations (Fig. 7a) suggests that Bonfioli had a relatively
fixed observation time, likely between 7:00 and 8:00 AM. This would explain the observed differences of 2–4°C in summer,
although certainly many other factors play a role (e.g., radiative biases). As best compromise we take 7:30 AM for the most
likely observation time. This value represents an average observation time as the actual time almost certainly varied slightly
from day to day (this uncertainty is covered in our error estimations, see Sect. 3.5).

In the afternoon (Fig. 7b) the differences are small, reflecting the slower temperature changes in that part of the day. The
best fitting observation times are between 3:00 and 4:00 PM. Therefore, our estimate is 3:30 PM.

As to whether the average observation times were the same throughout the record, the data indicate that Bonfioli probably
anticipated both morning and afternoon observations in the later years (especially from 1830, see Fig. 8) with the aim of
catching the daily minimum and maximum temperature. Therefore, we expect an inhomogeneity in the daily and monthly
mean temperature series, which will be dealt with in the next section.

The observation times given by Zallinger in 1809 (sunrise and 1 hour before sunset) are unlikely to have been followed in
the second part of the record given the differences shown in Fig. 7. On the other hand, they seem realistic for the early years
if we look at the average afternoon temperature for June–September (Fig. 8a), which are about 1°C lower during 1800–1809
than during 1820–1829 – compatible with a late afternoon measurement.

Assigning a time to sunrise and sunset at the bottom of an Alpine valley – where they can deviate strongly from their
astronomical equivalents – is not straightforward. We estimate them from the modern-day mean diurnal cycle: we take sunrise
as the earliest time of the day with a temperature increase larger than 0.2°C over a 15-minute interval, and sunset as the time
of the largest temperature decrease in the afternoon. This results in sunrise being between 6:15 AM in summer and 9:15 AM in
winter; and sunset between 6:45 PM and 4:15 PM, to which 1 hour needs to be subtracted. We assume that these observations
times were in place until April 1816, when twice daily measurements began.

**4.2  Temperature Accuracy in Bolzano**

The visual description of precipitation available for Bolzano allows us to assess the accuracy of the temperature values, at
least when they are close to 0°C. To do this we plot the frequency of snow entries with respect to all precipitation entries as
a function of temperature (Fig. 9). Jennings et al. (2019) give for the modern-day station of Bolzano a snowfall frequency of
50% at 0.5°C. We find the same frequency at about 1.5°C, suggesting a positive bias in the order of 1°C. Such a bias is very
common in early instrumental temperature records (e.g., Brugnara et al., 2022a) and is usually related to the exposure of the
thermometer.

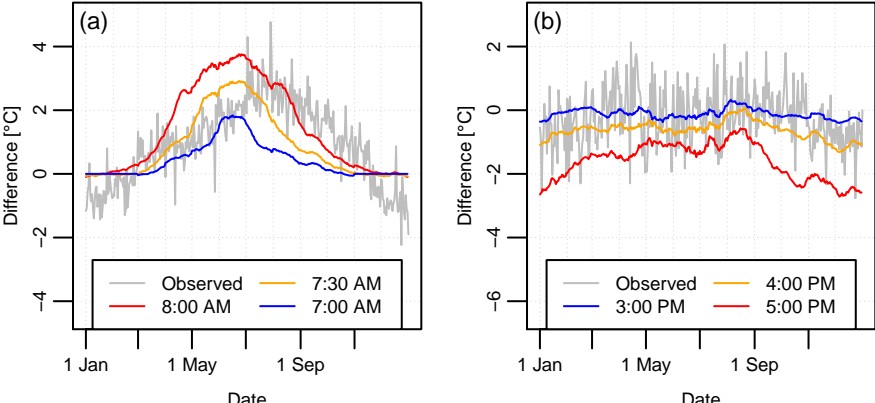

**Figure 7.** Mean temperature differences between Bonfioli and the anonymous observer in Rovereto during 1827–1829 for (a) morning and (b) afternoon observations. The colored lines represent expected differences – based on the modern-day mean diurnal cycle – for different assumed observation times in Bonfioli's record.

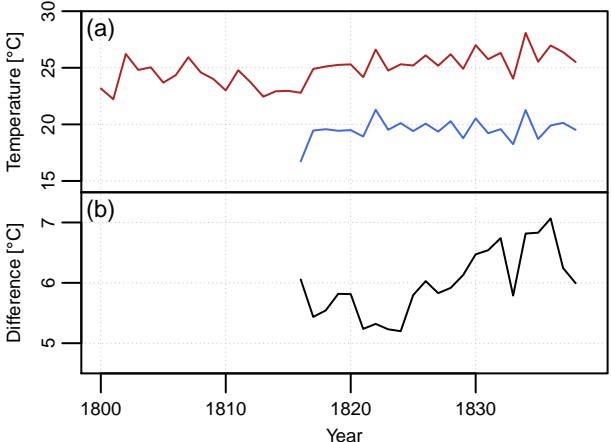

**Figure 8.** (a) Mean morning (blue) and afternoon (red) temperature during June–September in Rovereto. (b) Difference between the two series in (a).

However, this result should be interpreted with caution because the early precipitation observations might not be fully consistent with the modern ones. For instance, sleet and mixed-phase observations were excluded in the modern data but were classified as either snow or rain in the early data. Also the coarse resolution of the early temperature data complicates the comparison with the modern data, which were aggregated in $1°C$ bins (see Jennings et al., 2018).

If we look at the single observation times we find a much larger bias in the afternoon measurements, suggesting a radiative bias that might have been enhanced by radiation scattered by the snow on the ground. An alternative explanation is that the descriptive weather entries might have been a synthesis of the previous hours rather than instantaneous observations.


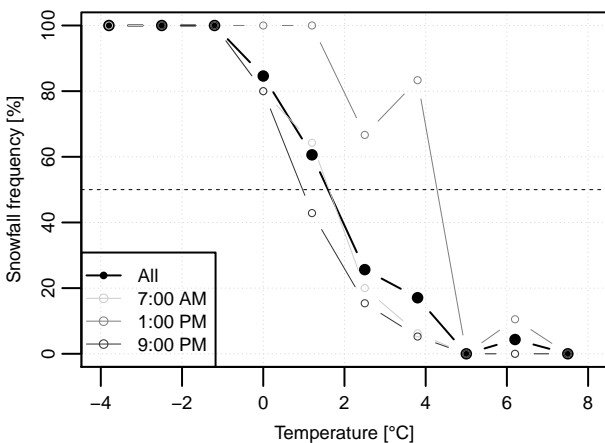

**Figure 9.** Snow frequency as a function of temperature in Bolzano. The horizontal line indicates the 50% level.

**Table 1.** Mean monthly corrections applied to the homogenized data series of Rovereto

| Parameter | Start | End | Jan | Feb | Mar | Apr | May | Jun | Jul | Aug | Sep | Oct | Nov | Dec |
|---|---|---|---|---|---|---|---|---|---|---|---|---|---|---|
| Pressure (hPa) | 1800-01-01 | 1814-12-31 | 0.0 | -0.8 | -1.3 | -1.4 | -1.1 | -1.0 | -1.1 | -1.3 | -1.2 | -0.6 | 0.1 | 0.4 |
| Temperature (°C) | 1800-01-01 | 1802-11-09 | -3.0 | -2.8 | -1.3 | 0.8 | 2.1 | 2.2 | 1.6 | 1.0 | 0.7 | 0.4 | -0.5 | -2.0 |
| | 1802-11-10 | 1808-11-23 | -2.7 | -2.5 | -2.0 | -1.5 | -0.8 | 0.0 | 0.8 | 1.0 | 0.6 | -0.4 | -1.6 | -2.4 |
| | 1808-11-24 | 1812-05-31 | -1.2 | -0.8 | -0.7 | -0.7 | -0.4 | 0.4 | 1.4 | 1.8 | 1.2 | 0.0 | -1.1 | -1.5 |
| | 1812-06-01 | 1816-04-20 | -1.5 | -1.2 | -1.0 | -0.9 | -0.9 | -0.5 | 0.2 | 0.7 | 0.7 | 0.1 | -0.8 | -1.4 |
| | 1829-06-01 | 1839-08-27 | -0.2 | 0.0 | 0.0 | -0.3 | -0.8 | -1.0 | -0.9 | -0.5 | -0.2 | -0.2 | -0.4 | -0.4 |

### 4.3 Homogenization and Comparisons with Validation Series

**4.3.1 Pressure**

Mean annual sea level pressure (SLP) anomalies are in good agreement with other stations in northern Italy and with the EKF400 reconstruction (Fig. 10). However, the first part of the Rovereto record is less homogeneous, as can be seen from the larger differences with the reference series. This may be related to a larger variability of the observation times.

To homogenize the pressure record of Rovereto we set one breakpoint at the end of 1814 that led to an average correction
of -0.8 hPa (Table 1). Smaller inhomogeneities remain but they are negligible for most applications. The pressure record of Bolzano did not require any correction.

The comparison of the daily mean SLP values of Rovereto with those of Milan, Padua, and 20CR (Fig. 11) confirms that the pressure measurements of Bonfioli are of remarkable quality for the time, despite the corrections that they required (i.e.,



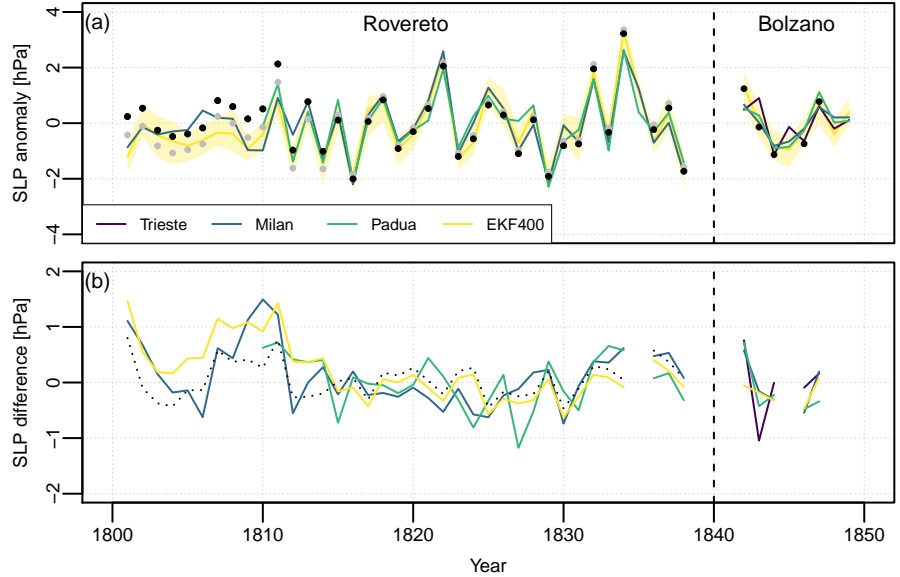

**Figure 10.** (a) Annual SLP anomalies in Rovereto and Bolzano (dots) and in the validation series (lines). The gray dots represent homogenized data, the shaded area represents the ensemble spread of EKF400. The reference period is 1810–1838 for the data before 1840 and 1842–1849 for the data after 1840. (b) Differences between the Rovereto and Bolzano annual anomaly series and the validation series. The dotted line represents the difference between the homogenized data and EKF400.

temperature and diurnal cycle correction). In particular, the correlation coefficients with the data from Milan and Padua are not

too far from those found in modern-day data (e.g., 0.98 for 1986–2000).

Our assumption that Bonfioli measured somewhere in the city center leads to a positive bias of ca. 3.5 hPa. Rovereto spans more than 50 m in elevation, therefore it is possible that Bonfioli measured at a lower elevation than assumed. In any case, early barometers were very likely to suffer from systematic biases (Brugnara et al., 2015). For instance, from the diameter of Bonfioli's barometer a negative bias of ca. 2 hPa should be expected because of capillarity (Camuffo et al., 2006). Moreover,

the bias has a seasonal component (see Fig. 6), perhaps because of exposure to solar radiation.

The pressure measurements from Bolzano are clearly of lower quality. Not only is their correlation with nearby stations lower, but they also suffer from a negative bias of the order of 10 hPa. Moreover, they seem to overestimate low pressure values. These problems suggest that a rather small cistern barometer or a siphon barometer was used (see Brugnara et al., 2015). The very coarse reporting resolution (ca. 3.5 hPa) also contributes to the low correlation coefficients.

The pressure data from 20CR have too little variability. This is at least in part related to the fact that we use the ensemble mean, but probably also because no data are assimilated in our study region. Note that the version of 20CR that we use begins in 1836, therefore the overlap with the Rovereto series is only 44 months.

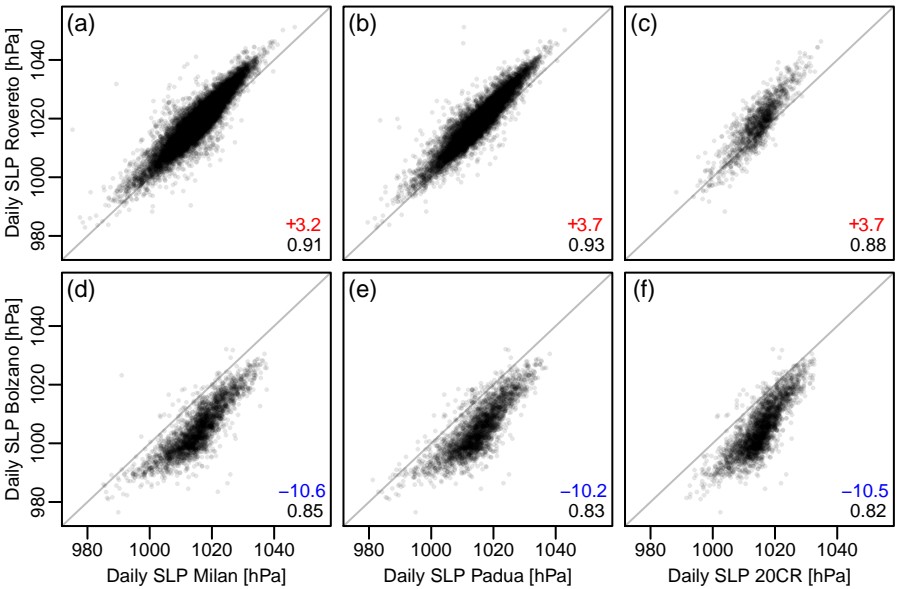

**Figure 11.** Homogenized daily SLP data of Rovereto (a–c) and Bozen (d–f) compared with Milan, Padua, and the nearest grid point in 20CR (ensemble mean). The colored numbers represent the mean difference in hPa, the black numbers the Pearson correlation coefficient.

### 4.3.2 Temperature

The temperature record of Rovereto have larger homogeneity issues than pressure (Fig. 12a), which can be expected from the
uncertainties in the observation times. We detected five breakpoints (see Table 1; breakpoints were chosen to match changes in observation time when applicable), four of which are in the part of the record that has only one daily measurement. In particular, winter temperatures until 1816 were too high (Fig. 13a), suggesting perhaps a too sheltered location of the thermometer. On the other hand, the record of Bolzano does not have clear inhomogeneities.

After homogenization the agreement with the reference series is comparable to that found for pressure (Fig. 12b). The "Year
Without a Summer" 1816 is prominently the coldest year in the covered period, while 1822 is the warmest year. As one would expect, the correlation of Rovereto's temperature with Trento (0.88) is higher than that with the Po Plain stations of Milan and Padua (0.82 and 0.78, respectively) over the common period 1816–1838.

Table 2 illustrates an additional problem affecting pre-1816 data: when comparing the standard deviation of daily mean temperature values with modern-day data we find too little variability in the earliest period in summer months. Since the sum-
mertime daily means of that period are based on a single observation in the afternoon, this again indicates that the thermometer was kept in a place where the amplitude of the diurnal temperature cycle was reduced with respect to modern measurement standards. Another factor that could contribute is an inaccurate estimation of the observation time.

It is important to remark that a small diurnal cycle does not imply poor quality of the measurements. For example, temperature data from the centennial observatory of Rovereto have been shown to have on average a diurnal range nearly $4°\mathrm{C}$ smaller



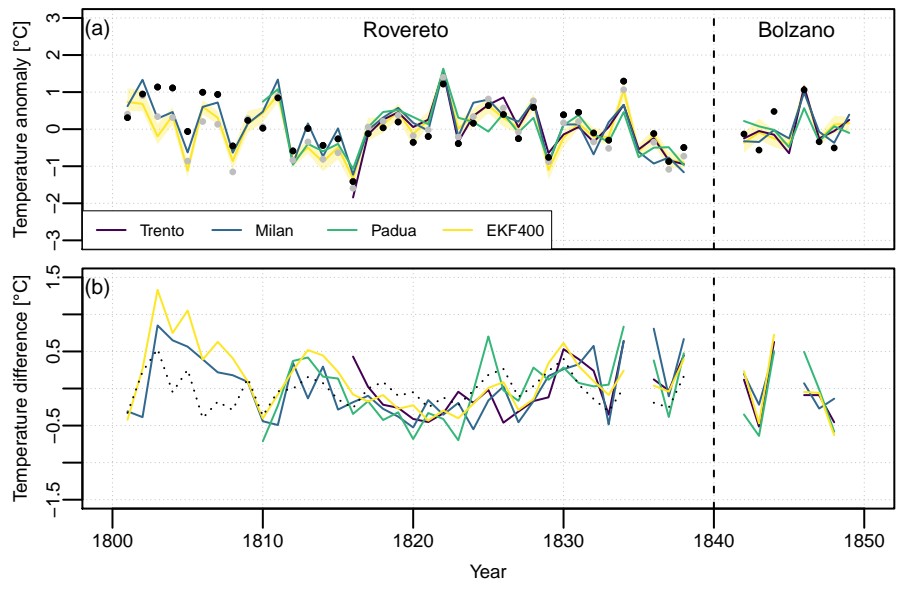

**Figure 12.** (a) Annual temperature anomalies in Rovereto and Bolzano (dots) and in the validation series (lines). The gray dots represent homogenized data, the shaded area represents the ensemble spread of EKF400. The reference period is 1816–1838 for the data before 1840 and 1842–1849 for the data after 1840. (b) Differences between the Rovereto and Bolzano annual anomaly series and the validation series. The dotted line represents the difference between the homogenized data and EKF400.

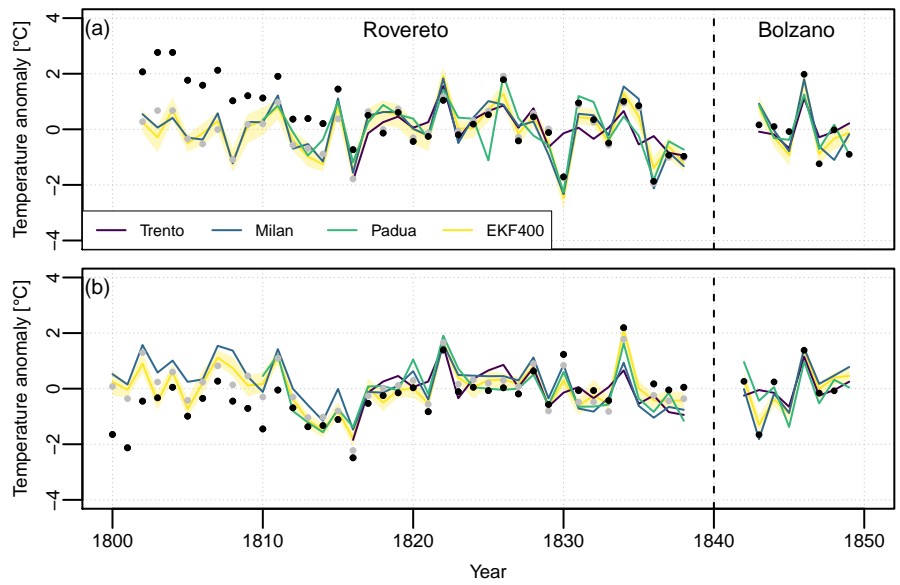

**Figure 13.** Same as Fig. 12a for (a) November–March and (b) May–September.

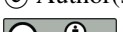

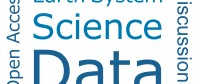

**Table 2.** Standard deviation of the homogenized daily mean temperatures (in °C) in Rovereto for each month in two sub-periods of the early instrumental record compared with modern-day data

| Month | 1800–1815 | 1816–1839 | 2011–2021 |
|---|---|---|---|
| Jan | 2.8 | 2.8 | 2.4 |
| Feb | 2.7 | 2.9 | 2.8 |
| Mar | 2.7 | 2.9 | 3.1 |
| Apr | 2.9 | 2.8 | 3.0 |
| May | 2.3 | 2.9 | 3.0 |
| Jun | 2.3 | 3.0 | 3.2 |
| Jul | 2.3 | 2.7 | 2.6 |
| Aug | 2.1 | 2.6 | 2.8 |
| Sep | 2.4 | 2.6 | 2.7 |
| Oct | 2.7 | 3.2 | 2.9 |
| Nov | 2.7 | 3.1 | 3.1 |
| Dec | 2.4 | 2.8 | 2.4 |

than the Meteotrentino station (Brugnara et al., 2016). Nevertheless, it negatively affects the daily means because we assume the modern-day diurnal cycle for their calculation.

## 4.4 Errors

The standard errors for the pressure and temperature series are summarized in Fig. 14 and 15, respectively. Clearly, the most influential parameter on the errors is the number of observations per day, which causes the errors for Bolzano to be generally smaller.

The total pressure error for Rovereto reaches a maximum in winter, when pressure variability is largest. Conversely, the temperature error has a minimum in early winter, because the diurnal range is small and the morning observation is more likely to have been made before sunrise. In late winter (February–March), however, the uncertainty in the observation times causes a peak in the error, as it does in October. This is because the average diurnal range is largest near the equinoxes and, therefore, temperature changes faster over time.

Before 1816 the total daily temperature errors are very large, ranging between 2–4°C. Monthly temperature averages are arguably more useful for that period as the errors are never larger than 0.7°C. Pressure data quality is less affected by the time uncertainty and the limited number of observations, so that the daily error is always around 2 hPa for Rovereto and 1 hPa for Bolzano, while monthly errors are below 0.5 hPa. However, systematic errors must be considered.





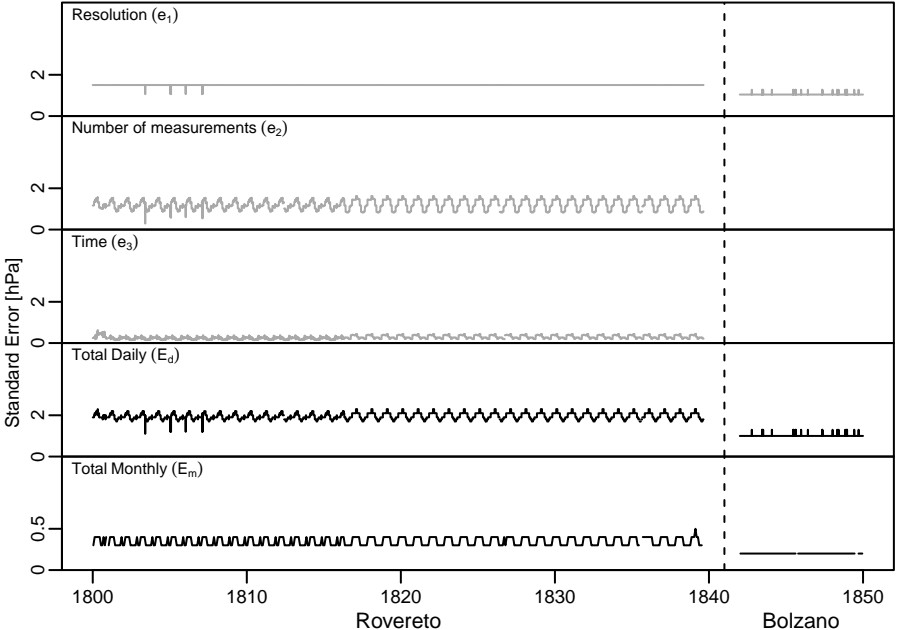

**Figure 14.** Components and totals of standard errors for pressure.

## 5 Assessment of Digitization Approaches

The performance of the OCR approach varied between ca. 87 % and ca. 97 % of correctly extracted data fields, with an average of ca. 93 % (Fig. 16). By contrast, the performance of the citizen science approach ranged between 98 % and 99.7 %, with an average of ca. 99 % correct fields. Both approaches had the worst performance for the year 1848 because of the low quality of many of the pages. For the same reason, about 10% of the data could not be extracted through the OCR procedure. Moreover, the year 1849 was not considered because of a significant change in the structure of the documents containing the data.

In general, the suboptimal quality of printing of the 1840s affected the performance of OCR. For more recent printed material we would expect a much narrower difference in performance between the two approaches. On the other hand, the performance of citizen scientists was higher than expected, being comparable to that of trained university students (Brugnara et al., 2020). However, as mentioned in Sect. 3.1.1, our results are probably too optimistic.

Another aspect to take into account is the quality of the errors: OCR produced errors that are in large part easy to predict and correct, whereas the errors made by citizen scientist are less predictable. In particular, about half of the OCR-related pressure errors are easy to detect because they produce values that are outside a realistic range. For temperature, the fraction is even larger when considering seasonality: for instance, the value "0" was sometime translated into "11" – a temperature that does not usually occur in winter. Therefore, in both OCR and citizen science approaches the error rate can be substantially reduced with simple solutions such as an automated quality check and multiple keying, respectively.



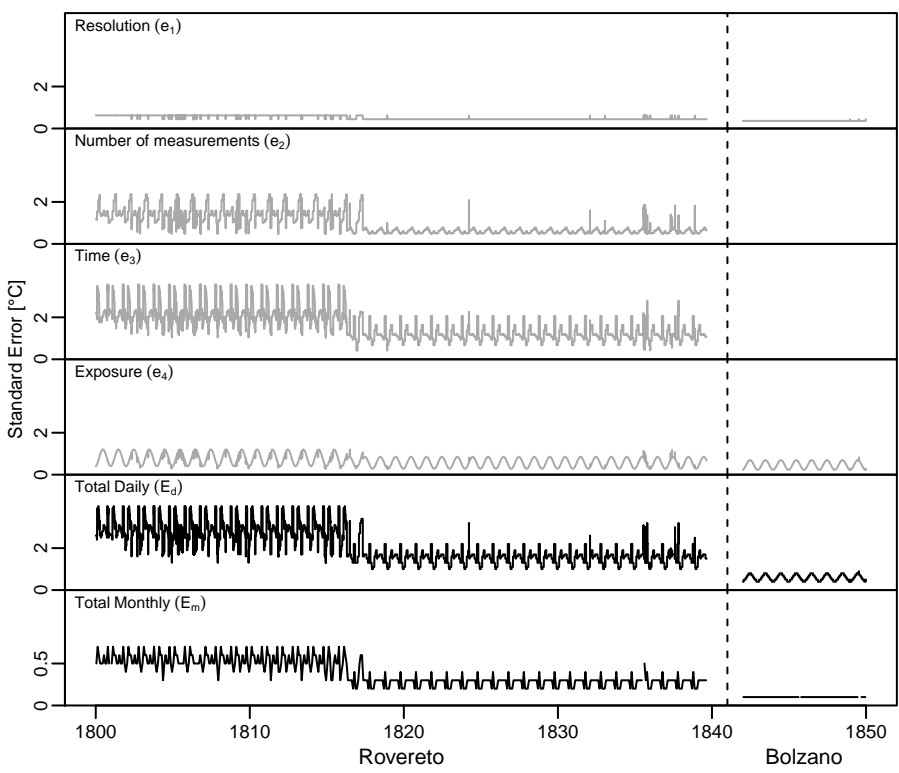

**Figure 15.** Components and totals of standard errors for temperature.

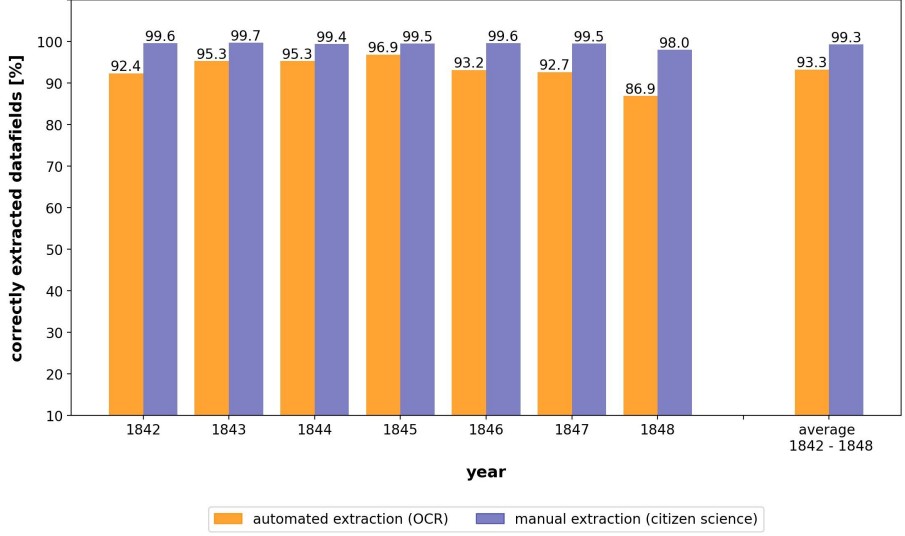

**Figure 16.** Percentage of correctly extracted data fields by year for the OCR and citizen science approaches.



## 6 Conclusions

We described two early instrumental records of pressure and temperature for the southeastern Alps covering the first half of the 19th century and representing the oldest data available for the region at daily resolution. In particular, we provide for the first time daily instrumental measurements in the region for the coldest period in the last 250 years – including the "Year Without a Summer" 1816.

Due to the lack of metadata many assumptions had to be made for the record of Rovereto, increasing the uncertainty particularly at daily scale. Even though the data have been homogenized, temperature data before 1816 should be used with awareness of the large errors attached to them, which we provide as quantitative estimates. In general, statistical approaches that can take errors into account (e.g., data assimilation) should be preferred when using the data.

We also compared different approaches to the digitization of the data for Bolzano from a contemporary newspaper. We found citizen science to give better results in terms of quality than a customized OCR algorithm. However, our results might not hold for more recent printed material of better quality. In general, OCR-based data extraction is worth being applied in situations in which the data to be extracted are well structured (constant table structure, constant printing type, high quality of printing and of scans) and the amount of data is very large. Citizen science offers a valid alternative in most situations and its performance can reach near perfection when requiring multiple typing (i.e., the same data point is digitized by different volunteers). Weather data published on newspapers are particularly suitable for citizen science because they are already fragmented in small chunks (a week of data or less), allowing a whole page to be easily transcribed by a single volunteer and thus greatly reducing the amount of work required to set up a data rescue project.

## 7 Code and data availability

The raw and processed data described in this paper are available at https://doi.org/10.1594/PANGAEA.946934 (Brugnara et al., 2022b). The raw data for Bolzano/Bozen in original units, including the weather descriptions, are provided as one comma-separated file. The raw data for both stations converted to modern units are provided in the C3S Station Exchange Format (SEF) (Brunet et al., 2020), consisting of one tab-separated file for each variable and each station. Each SEF file includes a header with standard metadata. The format is described in detail at https://datarescue.climate.copernicus.eu/node/80. The processed data (before and after homogenization) are provided as comma-separated files. The modern-day data for Rovereto can be obtained from the website of Meteotrentino (https://www.meteotrentino.it). The Python code for the OCR-based extraction is available through Zenodo at https://doi.org/10.5281/zenodo.7089123, and on the University Library Python Digital Toolbox (https://github.com/ub-unibe-ch/ds-pytools).

*Author contributions.* YB coordinated the digitization of the data for Rovereto, organized and promoted the citizen science project, collected metadata, and analyzed the data. MH wrote and applied the OCR-based extraction procedure, and keyed the data for Bolzano. IS carried out the archive work in Rovereto and photographed Bonfioli's weather diary. All authors contributed to writing the manuscript.



*Competing interests.* The authors declare no competing interests.

*Acknowledgements.* YB was funded by the European Commission (H2020; ERC grant no. 787574 PALAEO-RA). We are thankful to Nuria Plattner and the contributors of the University Library Python Digital Toolbox for providing help with code used in steps preparatory to the
335  OCR-based digitization. We also thank the volunteers who keyed the data on Zooniverse. Modern-day data for Rovereto were provided by Meteotrentino.



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
