# Peer review of "Two new early instrumental records of air pressure and temperature for the southern European Alps"

_Earth System Science Data, 2022_

## Author Comment (AC1)

We thank both referees for their feedback. Here we provide point by point answers (in *blue italic*) to their comments.

**REFEREE #1**

**General comment**

The recovery of past observations is always an important task, and so is this work. A significant part of the paper also deals with the data digitisation.

According to the title, the paper focusses on the time series, but a significant part of the article describes and discusses data digitisation (Sect. 3.1.1, 3.1.2 and 5). If digitisation is a relevant part of the work, I think that the title should mention it in some way, otherwise, the technical aspects could be moved to an appendix.

*We will move Sect. 3.1.1, 3.1.2, and 5 to an appendix.*

In any case, the paper appears unbalanced and its purpose is partly unclear. The Rovereto time series covers 40 years and was digitised by hand, while the Bolzano/Bozen time series only covers 8 years and was digitised with three different methods. The discussion about digitisation is only relevant for Bolzano/Bozen, which has a very short time series, despite the fact that data up to 1873 (24 more years) are available but they were not digitised (see line 170).

I have two questions: 1) Why the Bolzano/Bozen time series was not extended to 1873? 2) Do the authors want to illustrate the recovered historical data or the digitisation methods?

Perhaps the authors could consider either to complete the time series of Bolzano/Bozen or to drop it, and only discuss Rovereto.

*Unfortunately, the structure of the data tables of Bolzano changes a few times after 1849, requiring significant work on the OCR software and more resources than we currently have. Digitization using citizen science is ongoing but will probably require some years. We think that the early part of the Bolzano record is the most important because it is the only part that was not available in any form (not even as monthly means). The data from 1850 already exist as monthly means in the HISTALP dataset (Auer et al., 2007), at least for temperature. Therefore, our data will allow a seamless backward extension of the monthly Bolzano record.*

*The focus of the paper is on the recovered data. We will move the part on the digitization methods to an appendix, as suggested.*

A major issue is related to the amount and quality of metadata (e.g. instrument position and performance, observation times). They appear insufficient for the time series to be considered of high quality, therefore it seems that the observations described in the paper, particularly those of Bolzano/Bozen, have essentially a historical value. This point should be discussed in more details.

*This is a common characteristic of most early instrumental records, which are inherently of relatively low quality when compared to modern standards. However, there is still considerable scientific value in them. We will add a paragraph of this in the introduction and in the conclusions.*

Therefore, the paper cannot be published.

**Major points**

These are examples of insufficient metadata:

1. Lines 38, 43: Little metadata not provided by the diarist.
2. Lines 46, 49-50: Position of thermometer changed. Location of thermometer and observation times unknown.
3. Line 51: Barometer readings at unknown times.
4. Lines 114-117: Barometer elevation unknown.
5. Lines 246-249: 10 hPa roughly correspond to 100 m and there are many unresolved issues.

*We agree that the metadata is scarce, as it is for most early instrumental records. We will point this out better.*

**Specific points**

Line 82: Is the EKF400 time series suitable for homogenisation? Its horizontal resolution is 2°x2°and both Rovereto and Bolzano/Bozen lie in rather deep valleys surrounded by high mountains.

*This should not be a concern because long-term temperature and pressure variability has a large scale footprint (correlation in the Alpine region remains typically >0.8 even at 500 km distance, see Auer et al. (2007)). It would be a different story if we were dealing with, e.g., precipitation.*

Line 88: Does "self-keyed" correspond to "manual keying" (line 86)?

*Yes. We will rephrase that sentence.*

Lines 114-117: As the barometer elevation is unknown (see also lines 241-245), how was the reduction to MSL done? If elevation can be constrained within certain limits, the author could, in principle, estimate upper and lower limits of reduced pressure.

*We used the official town elevation, that is 204 m. This corresponds to the center of the old town and would also be a sensible educated guess. We will add this information to the text.*
*The upper and lower limits would make sense if the elevation would be the only source of bias; however, as mentioned in Sect. 4.3.1, there are likely larger biases affecting the measurements, such as the capillarity bias.*

Line 122: Please say explicitly that WMO (2018) is the source of the equations used for the data reduction.

*We will.*

Lines 126-127: Is that a reliable check? Moreover, in sect. 2.1 the authors say that the main weather events were also recorded in Rovereto. If possible one should use data of the same location. Please discuss the data availability in more details.

*This is a rather standard check done by many authors. We will give a few additional comparisons with results in the literature. Note that we do this check only for Bolzano.*
*The descriptive record of Rovereto is in the form of non-standardized narrative text (a couple of pages per month) and has not been digitized yet. We will add this information.*

Lines 139-140: I understand that eq. 2 holds for T1 and T2 separately and that T1/2 (line 140) means T1 or T2. If my interpretation is correct, I suggest to use "$T_k$, k=1,2" or something similar. If I am wrong, please explain it better.

*We will replace $T_{1/2}$ with "$T_1$ and $T_2$".*

Lines 142-152: This piece of text should appear at the end of sect. 3.4 because it logically follows the data quality control.

*We think that Sect. 3.3 should not be split, but we will merge it with Sect. 3.4 and rearrange the text to better represent the logical process: 1) quality control, 2) daily and monthly means calculation, 3) homogenization.*

Line 173: What is the "reporting resolution"? Is it related to the instrumental uncertainty? How much is it?

*This is the instrumental resolution inferred from the data. We will add more information.*

Line 184: "valid" instead of "non-missing".

*Will be changed.*

Line 189: The authors should discuss if the modern climatological daily cycle is really representative of that of the early 19[th] century. For instance, the soil characteristics at the station might be different, as well as the instrument exposure and elevation.

*It is certainly not ideal, but it is the best we can get. We will add a discussion on this.*

Lines 194-195: The sentence is unclear.

*We will rephrase it.*

Line 207: "lower" than what?

*We will rephrase.*

Lines 209-214: Could sunrise and sunset times be determined from direct observations in the towns of Rovereto and Bolzano/Bozen? The assessment based on temperature variations probably only holds on calm and clear-sky days. Please discuss the point.

*The timing of temperature changes in the climatology are very close to those of clear-sky days, because cloudy days have in general little temperature changes and thus have little influence. One could derive the times from solar radiation measurements instead: however, we found that the results are much more sensitive to the chosen threshold and that there is hardly a threshold/strategy that works well for all seasons. This is further complicated by temporary shadows from buildings and/or trees around the station. We will add a short discussion.*
*For Bolzano this is not relevant because the observing times are known there.*

Line 213: "4:15 PM and 6:45 PM" (reverse, in chronological order). Why has 1 hour to be subtracted?

*The observation time is one hour before sunset (see previous paragraph). We will add "to obtain the observation times" at the end of the sentence.*

Sect. 4.2: In the end, how accurate are the Bolzano/Bozen data? The uncertainties related to the method are not very encouraging …

*We will compare our results with those for other early instrumental records in the literature. In any case, one must keep in mind that the thermometer was most likely attached to a building (as it was common until the mid-20th century) and this usually brings a positive bias of the order of 1°C relative to a free-standing setting (see Brugnara et al., 2016). Therefore, a bias of 1°C would indicate rather high quality for the time. Unfortunately, we can only speculate on the reason for the larger bias in the afternoon.*

Lines 234-235: "average annual correction". Is the correction related to the uncertain instrument elevation?

*Most likely not. As shown in Table 1, the correction has a strong seasonal cycle, while for a change of elevation we would expect a more constant correction. This correction can probably be seen as an additional temperature correction; in other words, probably the relationship between outdoor temperature and temperature at the barometer - which we assumed identical - changed in 1814.*

Fig. 9: The line of 7:00 AM is faint.

Fig. 10: Panel a): The grey dots are rather faint. Panel b): Probably a dashed line is better visible than the dotted line (at least in the pdf).

Fig. 11: Panels a, b) The grey dots are rather faint.

*We will try to improve the figures.*

Line 284: Despite the last sentence, the paper misses the account of systematic errors.

*Systematic errors are discussed in Sect. 4.2 and 4.3. Estimates of the average errors are also given (Fig. 9, Fig. 11), except for the temperature in Rovereto. The next step would be to integrate the data into longer time series (e.g., extend the Bolzano temperature series in HISTALP backward to 1842), which would require an additional homogenization exercise to adjust the systematic errors. This is, however, out of the scope of this paper.*

*Auer, I., Böhm, R., Jurkovic, A., Lipa, W., Orlik, A., Potzmann, R., Schöner, W., Ungersböck, M., Matulla, C., Briffa, K. and Jones, P., 2007. HISTALP—historical instrumental climatological surface time series of the Greater Alpine Region. International Journal of Climatology, 27(1), pp.17-46.* https://doi.org/10.1002/joc.1377

*Brugnara, Y., Auchmann, R., Brönnimann, S., Bozzo, A., Berro, D.C. and Mercalli, L., 2016. Trends of mean and extreme temperature indices since 1874 at low-elevation sites in the Southern Alps. Journal of Geophysical Research, 121(7), pp.3304-3325.* http://dx.doi.org/10.1002/2015JD024582

**REFEREE #2**

Two early records of temperature and air pressure in two Italian cities were digitized and analyzed in this work. The quality of such records was evaluated, along with efforts conducted to homogenize and estimate the standard errors of these records. For one record (Bolzano/Bozen), two different digitization methods were used and the accuracy of these two methods were assessed.

While I found this work interesting and it is also important to obtain historical instrumental records, the objective and the methods applied in this work are confusing sometimes and discussions provided by the manuscript in its current form lack focus. Specifically, it is unclear whether the main objective of this work aimed to evaluate the obtained two historical records or to evaluate the two digitization methods. If the objective is to evaluate the two records with their observation times, data quality, and possible errors, it seems that the methods are not necessarily new and the methods from the previous work were often used and referenced without context of why these methods were used (e.g., the transformation of monthly corrections in Line 160 and error estimation using Brugnara et al. 2022a in Line 165). If the main objective is to compare the two digitization methods, the two approaches were only applied and assessed for half of the Bolzano record, which doesn't seem to be adequate. Assessing machine learning methods of digitizing the handwritten Rovereto records may be a more difficult task but likely provides more scientific merit.

The revision of the manuscript is therefore recommended to focus on one and more clear objective. If the objective is to provide and assess the two historical records, the paper should be more clear on why the selected methods such as for data quality checks, homogenization, and error estimation were used and aimed to achieve, what are the merit and limitations of using these two records for future studies, and how these two records contribute to the existing knowledge on historical Southern Alps climate. If the objective is to compare the digitization methods, additional digitization of the two records and comparisons especially on the Rovereto record are recommended. Some additional comments are provided below.

*We understand that the description of the digitization method created some confusion and we will therefore focus more on the data, by addressing the issues mentioned by the referee. Nevertheless, we will keep the part on the digitization in an appendix, as this might be interesting for some readers. Especially those who plan to start a digitization project could benefit from our experience.*

**Major comments:**

1. It may be useful to include a table listing all of the different historical records used and available in the region and the periods of these records. In addition to the two digitized records, the measurements from an anonymous observer mentioned in Line 189 and Milan, Padua, and 20CR data mentioned in Line 237 were also used for evaluation. Do such measurements (for example, the ones from the anonymous observer) are generally accepted to have a higher quality than the assessed two records?

   *We will add the table. The anonymous record was recovered by us and has never been assessed before. It has the advantage of providing three measurements per day at known times, which allows us to estimate the observing times of Bonfioli (we do not use it for validation purposes). Our assumption is that its quality is similar to that of Bonfioli's record.*

*The other records are published and widely used. Those of Milan and Padua, in particular, can be considered of high quality for the time. However, they are from quite distant locations.*

2. Similarly, a summarized table with a list of findings on the measurement time, accuracy, homogenization, and possible errors for the two time series of records would be useful to readers.

   *We will add the table.*

**Minor comments:**

1. Line 168: please provide additional details on the individual errors and why the equations such as equation (5) can be used to estimate such errors.

   *We will.*

2. Figure 7a: there seems to be a sudden increase of temperature difference around July 1$^{st}$, is there any reason why for such an increase?

   *No, that is likely just random noise.*

3. In Line 194, the "Bonfioli had a relatively fixed observation time" was summarized, while in Line 234, the reason listed is that the Rovereto record has "a larger variability of the observation times". It seems to be inconsistent between these two sentences.

   *The former statement refers to the period 1827-1829 (we then assume it to apply to the whole period after April 1816), the latter to the period until April 1816 (see lines 213-214).*

4. Line 238: measurements of Bonfioli are of "remarkable quality for the time", this is a subjective statement, needs some references or baselines to compare with.

   *It is hardly possible to find a baseline other than the given modern-time correlation, therefore we will remove the statement.*

5. Line 240: 0.98 as a correlation coefficient for 1986-2000, need a citation here.

   *This was calculated from the data at hand (Meteotrentino series and Milan/Padua series).*

6. Figures 12 and 13: it seems that the homogenization can largely improve the Rovereto record especially for the pre-1816 period, can a conclusion be drawn that the homogenized Rovereto record should be used?

   *We will add a paragraph on this in the conclusions.*